# Population Structure and Genetic Diversity of *Cucurbita moschata* Based on Genome-Wide High-Quality SNPs

**DOI:** 10.3390/plants10010056

**Published:** 2020-12-29

**Authors:** Hea-Young Lee, Siyoung Jang, Chea-Rin Yu, Byoung-Cheorl Kang, Joong-Hyoun Chin, Kihwan Song

**Affiliations:** 1Department of Bioresources Engineering, Sejong University, Seoul 05006, Korea; hylee@sejong.ac.kr (H.-Y.L.); chaery@sju.ac.kr (C.-R.Y.); jhchin@sejong.ac.kr (J.-H.C.); 2Department of Plant Science, Plant Genomics and Breeding Institute, College of Agriculture and Life Science, Seoul National University, Seoul 08826, Korea; hoifirstlove@snu.ac.kr (S.J.); bk54@snu.ac.kr (B.-C.K.)

**Keywords:** pumpkins, rootstock, bloomless cucumber, population structure, genetic diversity, core collection

## Abstract

Pumpkins (*Cucurbita moschata*) are one of the most important economic crops in genus *Cucurbita* worldwide. They are a popular food resource and an important rootstock resource for various Cucurbitaceae. Especially, *C. moschata* is widely used as a rootstock for the commercial production of bloomless cucumbers in East Asia. Since the genetic diversity of the commercial rootstock varieties is narrow, there has been an increasing demand for the trait development of abiotic and biotic stress tolerance breeding. In this study, 2071 high-quality SNPs that were distributed evenly across 20 chromosomes of pumpkins were discovered through the genotyping-by-sequencing (GBS) analysis of 610 accessions of *C. moschata* germplasm with a global origin. Using these SNPs, various analyses of the genetic diversity and the population structure were performed. Three subgroups were clustered from the germplasm collection, which included East Asia, Africa, and America, and these areas were included the most in each subgroup. Among those groups, accessions from Africa and South Asia showed the highest genetic diversity, which was followed by the Mexico accessions. This result reflected that large gene pools that consist of various native landraces have been conserved in those of countries. Based on the genetic diversity, we finally constructed the *C. moschata* core collection, which included 67 representative accessions from the 610 germplasms. Five morphological traits that are important in commercial grafting and rootstock seed production, which include the cotyledon length, the cotyledon width, the hypocotyl length, the internode length, and the number of female flowers, were investigated for three years and used to confirm the validity of the core collection selection. The results are expected to provide valuable information about the genetic structure of the worldwide *C. moschata* germplasm and help to create new gene pools to develop genetically diverse rootstock breeding materials.

## 1. Introduction

The genus *Cucurbita* (Cucurbitaceae) has five major domesticated species, which include *C. argyrosperma, C. ficifolia, C. maxima, C. moschata*, and *C. pepo,* which have been cultivated globally as an economic crop [1,2]. Among them, *C. moschata* is considered as the second most diverse species of the genus after *C. pepo*, and is closely related to *C. argyrosperma* in genomic status [2,3,4,5]. *C. moschata* exhibits remarkable morphological diversity in its fruit and seed types, and is cultivated and consumed in many parts of the world in forms of mature and immature fruit, seeds, young stems, tendrils, and flowers [6,7]. *C. moschata* is a versatile food source, and it has also been used as rootstock for vegetable grafting, particularly in Asia [6].

In most of East Asian countries, grafted seedlings are used routinely for commercial cucumber production for the purpose of overcoming continuous cropping disorders, escaping *Fusarium* wilt, overcoming the increase of temperature stress tolerance, and contributing to the production of high-quality fruit [8]. As cucumber stocks, fig leaf gourds (*C. ficifolia*), pumpkins (*C. moschata*), and Shintoza (*C. maxima* × *C. moschata*) are mainly used [9]. In Korea, fig leaf gourds are mainly used for cold season cultivation and pumpkins and Shintoza are generally used for warm season cultivation. Among them, pumpkins are used as bloomless rootstock, because they inhibit the formation of blooms on the fruit’s surface, which makes it possible to produce high quality cucumbers with a shiny fruit skin [10]. Bloomless rootstocks are particularly preferred in China even in the cold season, because the blooms are easily visible on the fruit’s skin of the long-type Chinese cucumbers, which have a uniform dark green color. In addition to the low-temperature tolerance, the increased resistance to diseases, pests, and the salt tolerance due to climate change and continuous cropping have been recognized as major breeding issues. However, since the genetic background of the current rootstock cultivars is narrow, it is necessary to develop gene pools for the breeding of new value-added varieties. However, it is required to understand the genetic structure and the diversity of the *C. moschata* germplasm for this.

There have been few studies about the genetic diversity of wild *C. moschata* germplasm until now, but there are many studies on the domesticated *Cucurbita* germplasm available, especially with *C. pepo* [11,12,13]. Even in the cases of the analysis of wild germplasm, the accessions consisted of small sizes of germplasm or landraces only from specific regions or countries [6,14,15]. One of the most recent studies is about the genetic variability among 91 accessions from different regions of Brazil that used agro-morphological characteristics, such as seed related traits and carotenoids contents. They clustered the germplasm into 16 groups according to the phenotypic variance and suggested the promising accessions by the per se approach for the effective application to breeding programs [15]. Another study showed the genetic divergence according to the elevation gradient in Mexico. They assessed the genetic data from 11 nuclear microsatellite loci and the mitochondrial locus of landraces from 24 localities [6]. As such, the studies to date have mainly focused on the analysis of genetic relationships based on specific genetic patterns or morphological variations for limited genetic resources with a narrow geographic origin.

In this study, we gathered a large number of *C. moschata* germplasm from various origins and assessed the diversity with high quality SNPs that were distributed evenly among the pumpkin genome, which had a purpose to understand the genetic diversity and the population structure. Furthermore, we selected the compact size of the representative entries as a core collection and validated the core collection using the morphological traits related to the grafting and the flowering. The *C. moschata* core collection constructed in this study is thought to be efficiently applied to generate new gene pools and the consequent practical rootstock breeding.

## 2. Results

### 2.1. The Genome-Wide SNP Identification among 20 Chromosomes in C. moschata

Based on genotyping-by-sequencing (GBS) method, around 1.5 billion reads were generated among 610 accessions, and more than 2 million reads per an accession were considered for alignment. In order to gather the efficient SNPs, the *C. moschata* var. *Rifu* version 1 was used for a reference genome. Following Lee [16] methods, 75,893 SNPs were filtered from the 1,267,422 raw SNPs, which have an SNP quality over 30 and a read depth over 3. Finally, 2071 SNPs were selected for the high-quality genotype after following criteria: minor allele frequency (MAF) > 0.05, SNP coverage > 0.6, and inbreeding coefficient (IF) > 0.8 (Appendix A). All the high quality SNPs were evenly distributed among the 20 pumpkin chromosomes, and 52 SNPs from the scaffolds within the unidentified chromosome were gathered in chromosome number 00 (Figure 1 and Appendix A). More than half of the accessions were homozygous, and the maximum heterozygosity value was 0.20 (Appendix A). Therefore, we confirmed that all of the accessions used in this study were highly fixed in the genotype that fit for the genetic diversity study.

### 2.2. The Population Structure and the Genetic Relatedness of the Germplasm Collection

Using 2071 high-quality SNPs, the population structure analysis was performed among the 610 accessions. After determining the best K (Appendix A), we could identify that there are 3 major clusters in this germplasm collection. According to the Q-values (Appendix A), which was almost half of the germplasm collection, 309 accessions belonged to cluster 1. Following that, 104 and 197 accessions were distinguished in cluster 2 and cluster 3, respectively (Figure 2a, Table 1).

The distribution of the germplasm origin showed that most of the accessions belonging to Africa were placed in cluster 2. The American accessions were mostly located in cluster 3, and more than 300 accessions from Asia were distributed in cluster 1. Only eight accessions were native to Europe, and seven of them were placed in cluster 3 (Table 1, Appendix A). Comparing the relationship between the genetic classification from the STRUCTURE results, and the origin of each accession, it was confirmed that each cluster corresponded roughly to the origins of the germplasm.

The genetic relatedness within the entire germplasm were shown in phylogenetic tree based on the UPGMA clustering method, resulting in seven subgroups (Figure 2a). Comparing this with the STRUCTURE results, group 1 and group 2 belonged to cluster 1 from the STRUCTURE analysis, group 3 was in cluster 2, and the remaining groups, which included group 4, group 5, group 6, and group 7 were in cluster 3 (Figure 2a). In terms of geographical distribution, group 1 and 2 are mainly composed of Korean landrace and commercial cultivars, respectively. Group 3 consists of accession from India and African countries. Group 4 mainly came from North and Central America, group 5 mainly came from South America, and group 6 mainly came from Mexico (Appendix A).

Among them, group 7 was clearly distinguished from the other groups, which is similar to the results from the PCA (Figure 2b), and the accessions that belonged to group 7 showed a slightly different phenotype in the species level. This might be a miss classification when the passport data was recorded. Therefore, 15 accessions that belong to group 7 were excluded in the further calculations for the genetic diversity and the statistical analysis.

### 2.3. The Genetic Variations of the Germplasm Collection

In order to understand the genetic diversity among the germplasm collection, the *F*-statistics were calculated from the 595 accessions without miss identification. The binary allelic data per locus was used for the statistical analysis and more than 1.3 alleles were effective. As expected, the heterozygosity (H_E_) and the Shannon’s diversity index (I) are the basic tools that are used to see the differences among the three clusters, and the average genetic diversity of the *C. moschata* germplasm collection was estimated to be 0.281 and 0.434 in the H_E_ and the I, respectively (Table 2). According to the results, cluster 2 showed the highest genetic variability (H_E_ = 0.327 and I = 0.494), whereas cluster 1 showed a lower genetic diversity than the others (H_E_ = 0.202 and I = 0.328). According to the inbreeding coefficient (F), all three clusters were estimated over 0.7, which reflected the high inbreeding within the cluster. Comparing the value of the H_O_ to the H_E_ in each cluster, we could also expect that all the clusters were forced to inbreed, which support the F values (Table 2).

The genetic difference among the three clusters was estimated using the population divergence (F_ST_). The calculated F_ST_ value of 0.48 indicated the largest difference between cluster 1 and cluster 3, whereas the smallest difference was found to be 0.21 between cluster 2 and cluster 3 (Figure 2c).

In order to define the patterns of genetic variation and to estimate the variance components among the sub-population, an analysis of molecular variance (AMOVA) was performed based on the pairwise genetic distances using GenAlEx 6.503. AMOVA revealed that 9.5 % of the total variation in the *C. moschata* germplasm was explained by the differences among the clusters, whereas 90.5% was explained by the differences within the clusters (Table 3). This confirms that there is great variation among the accessions within the germplasm.

### 2.4. The Construction of the C. moschata Core Collection and the Validation with the Phenotypic Variation

The main purpose of this study is to select the core accessions, which are composed of reasonable sizes that could represent the genotypic diversity of the entire germplasm. Taking this into account, 67 accessions were selected from the 595 accessions based on the genetic variation that was calculated from the 2071 SNPs that were evenly distributed among the pumpkin genome (Appendix A). All the core accessions were evenly distributed on the PCA plot, which suggests that the unbiased selection was made properly (Appendix A).

In order to validate the unbiased selection, various phenotypic data was used to compare the entire germplasm and the core accessions among three survey years. The traits used for the validation were the cotyledon length, the cotyledon width, the hypocotyl length, the internode length, and the number of female flowers in ten nodes. The former three seedling traits are important to produce high-quality graft seedlings, and the latter two are considered as important traits in the cultivation and the seed production.

The phenotypic range of the entire accessions for the cotyledon length was from 13.7 mm to 95.6 mm, which had an average 57.0 mm over the three survey years, whereas the core accessions ranged from 13.7 mm to 85.0 mm, which had an average of 52.4 mm over the three survey years (Table 4). The range for the cotyledon width was 8.8 mm to 58.5 mm, which had an average 36.7 mm in the entire germplasm, and 15.6 mm to 49.0 mm, which had an average of 34.8 mm, was the range in the core collection. With the hypocotyl length, two years of data except for 2018 was estimated to be 15.5 mm to 94.4 mm, which had an average of 44.6 mm, in the entire collection and 22.0 mm to 74.2 mm, which has an average of 41.0 mm in the core collection. The internode length was estimated by dividing the total length of 10 to 20 nodes by the node numbers in the range of internode length that were 2.4 cm to 30.3 cm in both of the entire collection and the core collection with an average of 15.6 cm and 16.3 cm, respectively. Both in the entire germplasm and the core collection, 0.2 flowers to 5.5 flowers were identified in ten nodes, which had an average of 1.2 flowers and 1.3 flowers in each collection. According to the distribution of the phenotypic range of the five traits, it was concluded that the accessions of the core collection covered most of the range of the phenotypic distribution.

## 3. Discussion

Pumpkins (*Cucurbita* spp.) are one of the vegetables that have great socio-economic importance. Many of the *Cucurbita* germplasms have been cultivated over generations from family-based farmers in many countries. The selection of these germplasms has been practiced over time by these populations, which is associated with the exchange of seeds between them, and the natural occurrence of the hybridization in the germplasm of this species has contributed to its increased variability [15].

Among the genus *Cucurbita*, we have focused on one of the major domesticated species, the *C. moschata*, which are cultivated globally and used for rootstock in the cucumber cultivation. Compared to other major crops, the breeding research on rootstock is still in the beginning stage worldwide, and the exploitation of useful genetic resources is still insufficient, so securing genetic resources that have agricultural value should be a top priority with this crop. In order to do this, we gathered a large number of germplasms that have a diverse origin from 42 countries and extracted the high quality 2071 SNPs throughout the 20 *C. moschata* chromosomes. It is a remarkable size of germplasm and genetic markers compared to the previous studies, in which only three morphological characters or 11 SSR markers that contained 122 alleles were used to assess the genetic diversity in Brazil or Mexico [6,15].

Among the 610 *C. moschata* germplasm used in this study, 15 accessions showed different morphology in the species level during the phenotypic evaluation, even though they are classified as *C. moschata* in the passport data. It is also common in other studies using genetic resources from various institutions and can occur due to misidentification in species level or by interspecific hybrid [17]. Therefore, to reduce the experimental errors, we confirmed also in the genetic differences of 15 accessions from both in the PCA and in phylogenetic analysis as a separate group. Finally, a total of 595 accessions, excluding 15 uncertain accessions, were used for further genetic analysis in this study.

According to the structure analysis in this study, three distinct clusters were identified within the worldwide *C. moschata* germplasm collection. The first cluster mainly consisted of East Asian accessions, which includes the biggest number of accessions. Additionally, it is divided into two sub-groups in the phylogenic tree, which group 1 showed a genetic homogenous that mostly consisted of Korean landrace, whereas group 2 included some Korean commercial cultivars that showed a genetic admixture. According to the values of the F_ST_ and the diversity index, this cluster showed the biggest difference to the others, and it had the smallest genetic diversity. These results could suggest that the germplasm in East Asia have a narrow genetic diversity, because they have been locally isolated and developed under independent breeding using limited genetic resources, even though it includes the largest number of accessions.

The second cluster consists of the African and the South Asian accessions, which apparently appear to be closely related to each other. And the last cluster included accessions mostly from the America continent where more than 63% of them were from Mexico. Even though it has not yet been clarified, Mexico is believed to be the origin of the *C. moschata* due to its archaeological clues and the presence of a various landrace that represents the high genetic diversity [18,19,20]. According to the previous studies, we expected that the last cluster would have the highest genetic diversity. However, based on the results of genetic analysis, it showed the biggest genetic diversity in the second cluster rather than the third cluster, that Mexico belongs to. Therefore, Africa and South Asian countries, especially India, were also thought to need to be carefully considered as the origin of *C. moschata*.

Based on the genetic diversity of the entire germplasm, we selected 67 representative accessions to establish the core collection of the *C. moschata* for the final purpose. In order to construct the core collection in this study, the large number of 2071 high-quality SNPs were considered, and phenotypic validation were also performed to ensure that we obtained a good representative.

The traits assessed in this study are important quality factors for the grafting, the cultural management, and the seed production, which can be considered as important targets in the rootstock breeding. The seedling characteristics of the rootstock, such as the small cotyledon and the short hypocotyl are taken into the primary target trait in the rootstock breeding. Since the main purpose of this study is to select the core collection that can be applied to the rootstock breeding through creating new gene pools, the above phenotype data would be practically applicable to choose the appropriate parental germplasm.

Actually, it takes a lot of time and effort to obtain consistent phenotypic data from a large number of germplasms of *C. moschata* because of their growth habits and the requirement of a large cultivation space. It could be explained that research that uses expressive phenotypic data obtained under the same environmental conditions is very rare [13,14]. Nevertheless, the phenotype evaluation in more than 600 accessions has been conducted over three years in this study, which was not only used as indicators in the selection of core collection but also would be useful for various further association studies. According to the results, the phenotypic distribution of the core collection mostly covered the entire germplasm variation, and it was thought that the core collection could represent the entire collection even in the level of the morphological diversity.

The large amount of genotypic and phenotypic data from this study would help to understand the genetic structure and the diversity of the *C. moschata,* and it could be practically applied in molecular breeding through further association studies.

## 4. Materials and Methods

### 4.1. Plant Materials and Phenotypic Evaluation

The total of 610 *C. moschata* accessions, which were originally from 42 countries, were collected from the National Agrobiodiversity Center Rural Development Administration (RDA) in Korea and the Agricultural Research Service (ARS) of United States Department of Agriculture (USDA). The scientific name and the origin of each accession were identified by referring to the passport data of the offering institute. However, some of them exhibited characteristics that differed from the general characteristics of the species. Additionally, in the genotype we identify genetic differences based on population structure, phylogenetic tree, and PCA. Therefore, it was excluded from the core collection construction. The number of SNPs and accessions used in each analysis were mentioned in the results section. The accession name, and the geographic locations are listed in Appendix A.

Over a three-year period (2018–2020), the entire germplasm collection was planted in plastic houses at Anseong-si, Republic of Korea, and three traits of seedling and two traits related with the seed production were evaluated. Phenotypes related to the seedlings, which included the cotyledon length, the cotyledon width, and the hypocotyl length, were investigated from the 10th to the 12th day after sowing using calipers. The internode length and the number of female flowers in ten nodes were counted on the 40th day after transplanting. All the traits were investigated with two replicates in two plants per an accession.

### 4.2. The DNA Extraction and the SNP Set Construction Using the GBS Method

The genomic DNA was extracted from the samples using the CTAB method [16], and they were diluted to 80 ng/µL in distilled water. The GBS library was conducted using the *EcoR*I/*Mse*I; enzyme set with an *EcoR*I; adaptor. The libraries were pooled in seven tubes. The contents of the tubes were sequenced in separate lanes using the HiSeq 2000 platform (Illumina, San Diego, CA, USA) at Macrogen, which is located in Seoul, Republic of Korea. All the 101 bp reads were trimmed using 96 different barcode sequences as previous described [21]. The filtered reads were aligned to the *C. moschata* var. *Rifu* (http://cucurbitgenomics.org/) as a reference genome sequence [22]. The reads mapping, the sorting, and the grouping were performed using BWA-MEM (0.7.12), SAMtools (1.1), and Picard (1.119). A GATK Unified Genotyper v.3.3-0 was used to call the SNPs [23]. The raw SNPs were filtered to remove the mono and the tri-allelic SNP types. The final SNP set was constructed after the filtering, which followed the following conditions. The minor allele frequency > 0.05, the SNP coverage > 0.6, and the inbreeding coefficient (IF) > 0.8.

### 4.3. The Genetic Analyses

In order to analyze the population structure of the entire germplasm (610 accessions) used in this study, we used a model based genetic clustering algorithm that is implemented in the STRUCTURE program ver. 2.3.4 [24]. The number of sub-populations (ΔK) was determined using the *ad-hoc* statistical method [25], which is based on the rate change in the log probability of the data between the successive K values. Ten independent runs for the K values that ranged from 1 to 15, were performed with a burn-in length of 50,000, which was followed by 50,000 iterations.

The phylogenetic trees were produced using genotyping data with 2071 SNP markers that used both the unweighted neighbor-joining method and the hierarchical clustering method based on the dissimilarity matrix calculated with the Manhattan index, which was implemented in the DARwin software (version 6.0.9). The entire germplasm was divided into subgroups based on the major nodes of the phylogenetic tree.

The *F*-statistics, which include the observed number of effective alleles (Ne), the heterozygosity (H_O_), the expected heterozygosity (H_E_), Shannon’s diversity index (I), the fixation index (F), and the genetic differentiation (F_ST_), were used for the analysis and the comparison of the diversity. Indices Ne, H_O_, H_E_, I, F, and F_ST_ were calculated using GenAlEx 6.502. The analysis of the molecular variance (AMOVA) was conducted in order to detect the genetic variance within and among the population using GenAlEx ver 6.502.

### 4.4. The Core Collection Selection and the Evaluation

In order to establish a core collection, the representative entries were selected based on the genotypic diversity analysis method of Lee [17]. The representative accessions were selected based on the advanced M strategy using a modified heuristic algorithm implemented in the PowerCore software version 1.0 [26]. In order evaluate the efficiency of the core entries from the whole accessions, five diverse traits were used to compare them. The phenotype data is presented as the mean ± SE.

## Figures and Tables

**Figure 1 plants-10-00056-f001:**
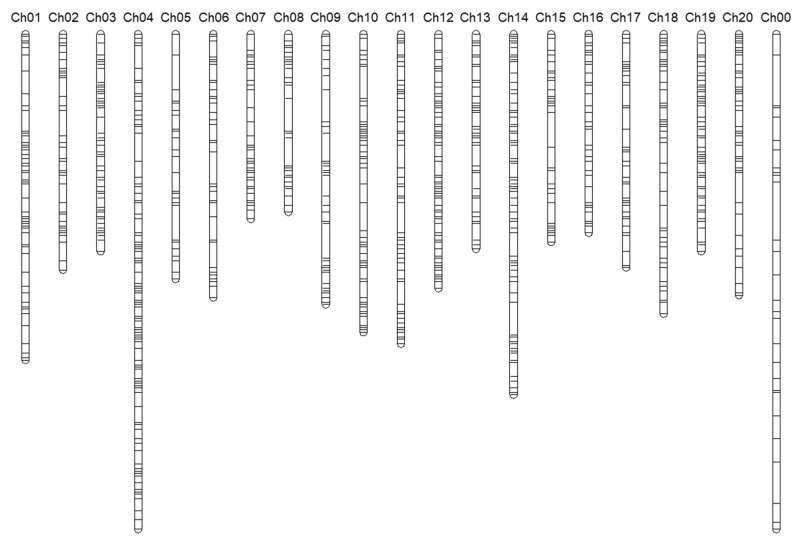
The genome-wide SNP distribution (2071 SNPs) in the *C. moschata* chromosomes.

**Figure 2 plants-10-00056-f002:**
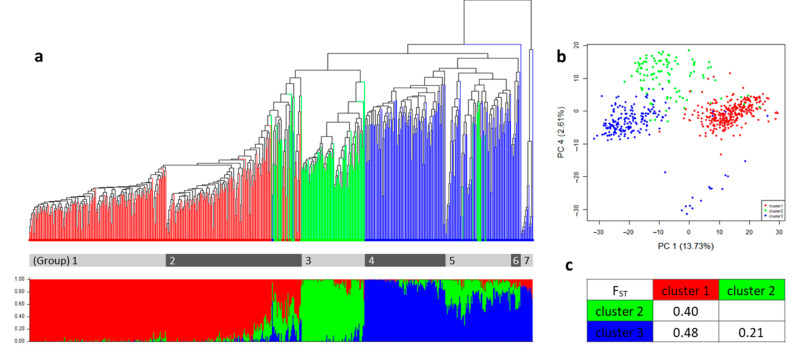
The population structure and the genetic relationship among the three clusters in the *C. moschata* germplasm collection. (**a**) The upper dendrogram was produced using the UPGMA method based on the genetic dissimilarity among the 610 germplasm accessions. Seven groups from the entire collection were distinguished in the gray color bar under the tree. The bar plots from the STRUCTURE analysis were listed in the order that is the same as the dendrogram under the tree. The Q-values of each clusters per accession were distinguished with different colors. Cluster 1 is in red, cluster 2 is in green, and cluster 3 is in blue. (**b**) The PCA of the entire collection colored by the clusters. (**c**) The F_ST_ values among the three clusters.

**Table 1 plants-10-00056-t001:** Origin of the *C. moschata* germplasm collection among the three clusters inferred by the STRUCTURE analysis.

Continent	Cluster_1	Cluster_2	Cluster_3
Africa	1	41	4
America	3	5	162
Asia	305	57	24
Europe	0	1	7
Total	309	104	197

**Table 2 plants-10-00056-t002:** The genetic diversity based on the genome wide SNPs among the three clusters.

Cluster	Number of Accessions	Na	Ne	H_O_	H_E_	I	F
1	309	1.985	1.311	0.045	0.202	0.328	0.756
2	104	1.994	1.549	0.094	0.327	0.494	0.701
3	182	1.998	1.526	0.068	0.315	0.48	0.777
Total	595	1.992	1.462	0.069	0.281	0.434	0.745

Na: number of different alleles, Ne: number of effective alleles, H_O_: observed heterozygosity, H_E_: expected heterozygosity, I: Shannon’s diversity index, F: inbreeding coefficient.

**Table 3 plants-10-00056-t003:** The analysis of the molecular variance (AMOVA) among the three types of subpopulations based on the genome wide SNPs.

Source of Variation	Degrees of Freedom (df)	Sum of Squares	Mean Sum of Squares	Estimated Variance	Percentage of Variation (%)
Among Pops	2	80,683	40,342	212	9.50%
Within Pops	592	1,202,784	2032	2032	90.50%
Total	594	1,283,468		2244	100%

**Table 4 plants-10-00056-t004:** A comparison of the phenotypic distributions between the whole germplasm collection and the core collection.

Traits	Type	Unit	2018				2019				2020			
			Min	Max	Mean	SE	Min	Max	Mean	SE	Min	Max	Mean	SE
Cotyledon length	W	mm	16.1	78.7	55.3	0.3	13.7	65.0	46.4	0.8	28.3	95.6	69.3	0.7
	CC	mm	27.8	64.5	52.3	0.9	13.7	53.7	42.3	3.0	48.0	85.0	62.7	2.1
Cotyledon width	W	mm	8.8	53.1	36.3	0.2	10.8	50.5	31.3	0.5	23.7	58.5	42.5	0.4
	CC	mm	23.4	45.5	34.3	0.6	15.6	35.8	30.1	1.6	31.3	49.0	40.0	1.0
Hypocotyl length	W	mm	-	-	-	-	15.5	61.7	35.2	0.7	22.0	94.4	54.1	0.9
	CC	mm	-	-	-	-	23.6	48.0	36.1	1.8	22.0	74.2	45.9	2.3
Internode length	W	cm	2.4	24.8	13.5	0.1	6.8	26.8	15.9	0.2	4.0	30.3	17.4	0.2
	CC	cm	2.4	24.4	14.2	0.4	7.0	26.8	15.9	0.6	11.2	30.3	18.7	0.8
No. of female flower among 10 nodes	W	ea	1.0	5.0	1.8	0.1	0.2	5.5	1.1	0.1	0.2	3.8	0.7	0.0
	CC	ea	1.0	4.0	1.8	0.2	0.4	5.5	1.7	0.5	0.2	1.0	0.6	0.1

W: the entire collection, CC: the core collection, SE: the standard error.

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
