# Peer review of "Population Structure and Genetic Diversity of Cucurbita moschata Based on Genome-Wide High-Quality SNPs"

_plants, 2020, doi:10.3390/plants10010056_

Round 1
Reviewer 1 Report
This manuscript identified high quality SNP markers using GBS analysis with 610 accessions of C. moschata and conducted population structure and genetic diversity analysis. Also, they developed core collection with 67 accessions for future breeding programs. With some minor modification, it can be acceptable in the journal of Plants.
- for better understanding, description for genomic status in C. moschata need to be in introduction.
- In this study, 67 core collection was created for C. moschata. However, I don't know the 67 accessions were good enough for working as core collection.
Author Response
Dear Reviewer,
Thank you for your time and consideration for our manuscript. Based on your comments, the answers to each question were answered in blue and the changes made in the revised manuscript were marked with highlighted text.

Reviewer 2 Report
This manuscript is very interesting and pertinent to improve the knowledge of pumpkins germplasm using high quality molecular method. Introduction is well construct and complete. Method and results sections could be improving principally with more explanations. Discussion section need really to be improve; indeed, some paragraphs are more results than discussion, and generally authors didn’t discuss their genetic results.
Keyword: avoid repeat word of the title as for example: curcubita moschata, genetic diversity.
Methods
Line 269: USDA ARS. Give a complete name
Line 272: I didn’t see the “management number” in the Table S1. Eliminate in the text or complete Table.
Line 270-271: “However, some of them exhibited characteristics that differed from the general characteristics of the species, so they were excluded from data analysis.”. It is not so clear. If some of them were excluded from data, how many authors kept for analyses, I suppose less than 610. But in Table S1, authors present 610 accessions. Please clarify.
Line 296: Authors used the reference of Evanno (number 23) for Structure program. It is an error. Evanno need to be cited for the method to determine the optimal K (Delta K) using the website. While Structure program have other references: Pritchard et al. 2000; Falush et al. 2007; and Hubisz et al. 2009.
Line 298: “Ten independent runs, for the K values that ranged from 1 to 15, were performed …” complete with comas
Line 293 and 304: the use of subtitles is confused. The section in the subtitle 4.3 is not the genetic diversity, but genetic differentiation methods to determine the genetic structure. And the section in the subtitle 4.4 is in fact the genetic diversity parameters (e.g., Ho, He, …), less Fst which is used to evaluate the genetic differentiation. To avoid confusion, I suggest used only one subtitle as for example: Genetic analyses. Also at the begin of genetic analyses, it would be better to inform about the number of SNP used for analyses. I suppose that it is the same file for all analyses.
Results
Line 84: I don’t understand the reference to Table S1 in this case because authors talk about reads obtained for the 610 accessions, but Table S1 didn’t have no information about reads, only the list of the 610 and genetic structure information (clusters). Please, clarify.
Line 86: “The SNP set construction was performed following Lee [16] methods.”. This is for methods.
Line 92-93: The Figure S1 shows that more than 50% of the accessions have no heterozygosity (value 0) but authors mention in the text “less than 0.05”, this is not the same. Also, authors mention that the maximum value is 0.95, but in the Figure we can observe the final value at 0.20. Also, error in the x-axis name. Adjust graph or correct in the text.
Figure 1: may be complete the legend with “…. In the C. moschata chromosomes”.
Line 101: “which was based on the ad-hoc statistical method”. This is method, eliminate.
Line 102: why said “confirm”. Authors identify these 3 major clusters or they confirm them comparing with other study? Clarify.
Structure analysis: (1) Give in supplementary material the graph based on the mean value of Ln P(D) for each value of K to show the determination of the best K, (2) give in supplementary material a table with the Membership probabilities (Q) of each pre-defined “groups” (7) in each of the new clusters (3).
Line 116-117: C. moschata in the legend of Fig. 2 need to put in italic.
Line 122: FST, need to put “ST” in subindice
Line 125-126: “The genetic relatedness within the whole germplasm was estimated using the genetic dissimilarity”. This is methods. Eliminate and adjust the next sentence if necessary.
Line 128-131: we have no idea what is the seven groups. It is necessary to explain in the method section this information.
Line 141: authors mention “populations” I suppose that refer to the 3 clusters determine by Structure analysis. It is not evident, please clarify.
Line 145: The fixation index in population genetic correspond to the FST value, not the same than inbreeding coefficient (F). Adjust. Also in the legend of Table 2.
Line 173-177: This is methods
Discussion
Line 208: originality? I suppose is: origin
Lines 213-218: this is not discussion, and it is a repeat section of the results section (lines 130-135). Eliminate.
Line 219: Why talk about “subpopulations” if before authors mention populations to talk about the 3 clusters identified by Structure analysis. Clarify.
Line 219: “divided” replace by “identify”
Line 223: Authors didn’t mention the origin of the different groups in the methods and results. We are learning in the discussion that group 1 correspond to Korean landrace, etc… It is necessary give this information before.
Line 229: we didn’t know which groups correspond to Africa and South Asian. It is necessary to give this information in methods and results sections.
Lines 229-236: At the beginning, we think that authors talk about African and South Asian, but in the same paragraph they talk about Mexico, and introduce a new term of “subgroup” no define before. It is really very confuse. As mentioned before, it is really necessary to present a table with information about origin of the 601 classified considering population, subpopulation, cluster, subgroup (how many successions in each category) … I’m really lost.
Line 237-238: In this sentence, authors used for the same information “subpopulation” and “population”. It is important to homogenize the term used to define categories.
Line 237-240: It looks more like results than discussion. Rewrite to discuss this result.
Line 241-245: this part is a repetition of text from results and methods. This is not a discussion.
Author Response

(The authors gave the same response as above.)
